# Adaptation of a Traditional Irrigation System of Micro-Plots to Smart Agri Development: A Case Study in Murcia (Spain)

**Jesús Chazarra-Zapata** [1] , **Dolores Parras-Burgos** [2] , **Carlos Arteaga** [3] , **Antonio Ruiz-Canales** [1,*] and **José Miguel Molina-Martínez** [4]

1  Engineering Department, Miguel Hernández University of Elche, 03312 Orihuela, Spain; jesuschazarra@gmail.com
2  Structures, Construction and Graphic Expression Department, Universidad Politécnica of Cartagena, 30202 Cartagena, Spain; dolores.parras@upct.es
3  Hidroconta S.A, I+D+I Company, 30012 Murcia, Spain; carlos.arteaga@hidroconta.com
4  Agromotic Engineering and the Sea R+D+i Research Group, Universidad Politécnica of Cartagena, 30202 Cartagena, Spain; josem.molina@upct.es
*  Correspondence: acanales@umh.es; Tel.: +34-649581413

**Abstract:** Currently, water users associations (WUAs) in semi-arid areas of southeastern Spain (Murcia region) send a multitude of data supplied by sensors in the field to the cloud. The constant technological revolution offers opportunities for small farms not to be abandoned, thanks to the Internet of Things (IoT). This technology allows them to continue to manage remotely using smartphones/tablets/laptops. This new system contributes to the mitigation of climate change from several aspects: reduction of water footprint and energy consumption (in the pumps that pressurize the grid, such as in the optimization of the proposed solution, by using batteries that communicate in low radiation of electric and magnetic alternating fields (LoRad), General Packet Radio Service (GPRS), or narrowband IoT (NB-IoT), or clean energy). The analysis of these data and the incorporation of new IoT technologies facilitate the maintenance of green roofs and ensure the continuity of these farms. The direct benefit obtained is remarkable $CO_2$ removal that prevents desertification by the abandonment of arable land. This communication shows the implementation of a Smart Agri system in areas with micro-plots (surface less than 0.5 ha) with low-cost technology based on long-range (LoRa) systems, easily maintainable by personnel with basic knowledge of automation, which transforms into a very interesting solution for regions with development roads. In addition, complex orography and difficult access are added in both physical and technological environments. The main technical limitations found in such plots are poor coverage for mobile phones and unworkable and expensive implementation by wiring or WiFi/radio systems. Currently, thanks to the Smart Agri system implemented in this WUA in Murcia, farmers can manage and control the irrigation systems in their plots from home. Then, they cannot lose their crops and respect the isolation conditions imposed by the Spanish government as a result of the alarm caused by COVID-19.

**Keywords:** carbon footprint; water footprint; LoRa; IoT; desertification; COVID-19

## 1. Introduction

The use of the Internet of Things (IoT) by water users associations (WUAs) contributes to the mitigation of climate change from several aspects, including the reduction of the water footprint [1–3] and energy consumption. Several wireless communication systems have been employed in these systems, such as long-range (LoRa) [4], among others. Maintaining green roofs and the continuity of

these farms can be done by incorporating new IoT technologies [5–7]. One of the direct benefits obtained is remarkable $CO_2$ removal [8], which prevents desertification by the abandonment of arable land. This paper shows the implementation of a Smart Agri system in areas with micro-plots (surface less than 0.5 ha) with low-cost technology based on LoRa systems [9]. In addition to the advantages mentioned above, these systems allow the remote management of fields with isolation conditions because of the restrictions caused by COVID-19 all over the world [10].

A case study located in Pliego (Spain) will be described. Pliego is a municipality in the central area of the region of Murcia. The Pliego water users association (WUA), due to the current drought that is hitting the east of Spain, increased energy consumption and its price, and the scarce availability of water resources for the irrigation of land, needs to act so that the traditional crops of farming families that are managed by descendants with new technologies do not become a desert. Forced to move to cities with new job prospects in order to survive the low yield produced by inherited smallholdings, these families have little time to pay attention to their farmland and represent the last link to the land of their parents. It is necessary for them to reduce and optimize the water demand for the irrigation of crops compatible with the terrain and to use new technologies to allow the management of their crops, dedicating a few minutes a day without having to travel to their farms. This is combined with elderly farmers who still manage the land in situ, looking skeptically at smartphones. The new technologies are applied in rural areas of mainly agricultural regions with a great history of technological development, as is the case in Murcia. They are the hope for certain types of micro-farms, which are not economically profitable, in order to avoid desertification (Sustainable Development Goal (SDG) 15.3: "By 2030, combat desertification, restore degraded land and soil, including land affected by desertification, drought and floods, and strive to achieve a land degradation-neutral world") [11–13]. These new technologies are capable of reducing some areas of water scarcity by abandonment. At the same time, they promote the sequestration of $CO_2$ by means of agricultural plantations in order to reduce the water footprint (SDG 12.4: "By 2020, achieve environmentally sound management of chemicals and all wastes throughout their life cycle, in accordance with agreed international frameworks, and significantly reduce their release to air, water and soil in order to minimize their adverse impacts on human health and the environment") [14,15]. Specifically, thanks to the evolution of IoT, there are combined solutions for these purposes. For example, narrowband IoT (NB-IoT) provides a technical solution, which, combined with the use and dissemination of smartphones, allows the rural population to fix problems. Moreover, this technology allows access by people located outside of the rural plots. Most of them are successors of farmers who have been forced to emigrate to cities in search of better job opportunities. This situation is forcing the abandonment of micro-plots. The remote control, automation, and management of irrigation systems of these crops (Figure 1) represent an opportunity to change this situation. Computer applications and energy-efficient software will give the opportunity to modernize and manage plots with minimum investment.

Annually, thousands of engineers from all over the world visit this region to learn about these systems, from high distribution networks (pumps, reservoirs, filters, among others) to irrigation systems close to crops (drip irrigation [16–19] collection and reuse of filtered water to the subsurface [20–22] and spraying [23], among others [24–26]).

In this scenario, more than 20 years ago [27], the first attempts to automate WUAs emerged. The intention was to obtain meter readings and remote control of the opening and closing of valves. Initially, the most sophisticated devices were used with Fieldbus technologies with two-wire power and communications. Separate communication power was included in other facilities (four wires). These early attempts failed due to the high cost of installation and wire network maintenance. In this case, it was the precise installation of miles of buried wire along hectares of farmed land. An alternative to wire networks was radio modem-based systems [4]. They consisted of integrated technologies from industry, adapted for use in rural environments because of their high consumption and sensitivity to weather factors. Deployment was easier due to wireless communication, but there was still a

high maintenance cost. This involves, on the one hand, the use of licensed communication bands, and, on the other hand, huge energy consumption.

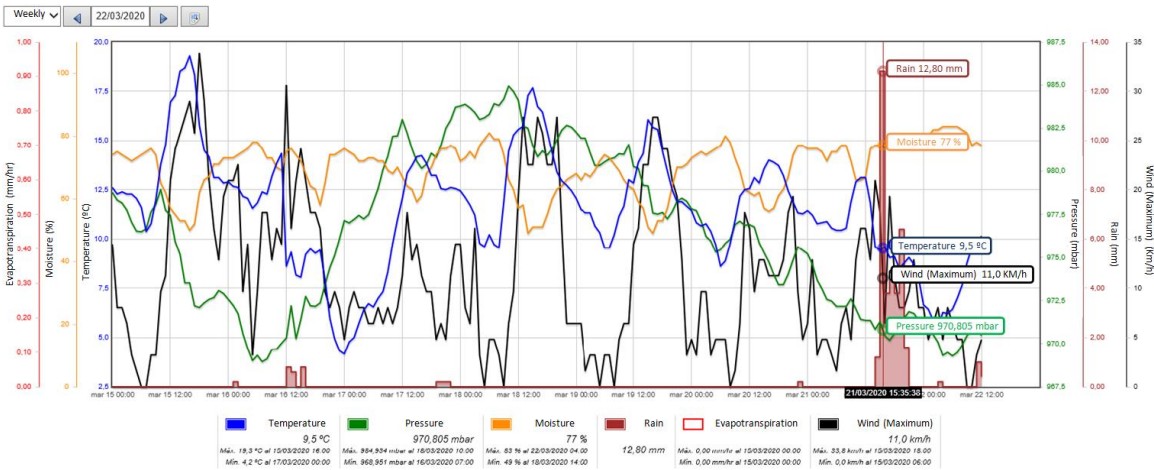

**Figure 1.** Graph of data available to users via the Internet of Things (IoT).

Energy consumption requires the installation of large solar panels. In addition, high power dissipation caused numerous breakdowns because of the overheating of equipment. With the improvement of wireless technologies (WiFi, Zigbee, etc.), communication modules based on free-band 433 MHz and then 868 MHz appeared [28,29]. They were quickly adapted to replace the old radio modems. With much lower acquisition and deployment cost, the biggest problem was the short communication distance. This situation forced the use of numerous hubs and repeaters in the network. This ultimately led to the development of less robust topologies due to the abundance of nodes. For hubs, however, radio modems were still used with the same problems described above. With the development of mobile telephony, the first equipment based on Global System for Mobile Communication (GSM) and later General Packet Radio Service (GPRS) began to appear [30]. This equipment, already designed specifically as solutions to be applied to irrigation, were much more adaptable and their deployment, with exclusive dependence on the coverage of the area, was very simple. As fully independent equipment transmitting directly to the Internet, they form extremely robust topologies. The aim is to design a system that allows the management and control of micro-plots operated through IoT technologies, helped by friendly apps that are manageable, even for people with basic knowledge about these technologies. All the technical problems of this specific case would be solved. At the same time, it would allow carrying out the procedures at any time of the day and anywhere in the world where there is an Internet connection [31–34].

## 2. Materials and Methods

The case study of the application of information and communication technology (ICT) in the Pliego WUA irrigation system includes several levels of devices and sensors. The main objective of this research was to integrate two types of systems: a new one in Sector 2 with larger plots, whose new implementation allowed the use of commercial hardware but required a minimum amount of control for 16 plots to be economically viable, compared to another for Sector 1 of smaller size and with more abrupt orography, capable of serving micro-plots with independent remote transmission units.

The main scheme of the system is described in Figure 2. The data provided by the different sensors are recorded according to their degree of importance (the greater the importance, the greater the frequency of reading), and are turned over to the remote transmission units (RTUs) for storage up to 6 months. When the communication ports are opened, they launch a LoRa signal that is picked up by the nearest concentrator, and this identifies the communication station and labels the date, compares whether the received data are more recent than what is stored and whether this is how it records them

(otherwise it does not store them), and this process is repeated to at least 2 concentrators to guarantee communication. In turn, if these concentrators have new instructions that have been sent from the app by users or administrators, they return them during the temporary communication window (repeating the verification process) and can thus reprogram the RTUs to modify the programming (such as opening and closing valves, increasing the frequency of meter reading, determining if an anomaly has been detected). Finally, after closing the communication cycle, which in our case was limited every 2 h to extend the battery life (it can be modified if there are problems), the data are uploaded to the network and recorded on a physical hard disk and optionally in the cloud, to later be viewed on the users' app and the supervisory control and data acquisition (SCADA) system. This does not mean that there is no update of the scale in 2 h, but all data provided by the sensors of all WUAs of the system are completed every 2 h. At the same time, critical points (primary network and some in the secondary) are read in real time and updated instantly in the app and the SCADA.

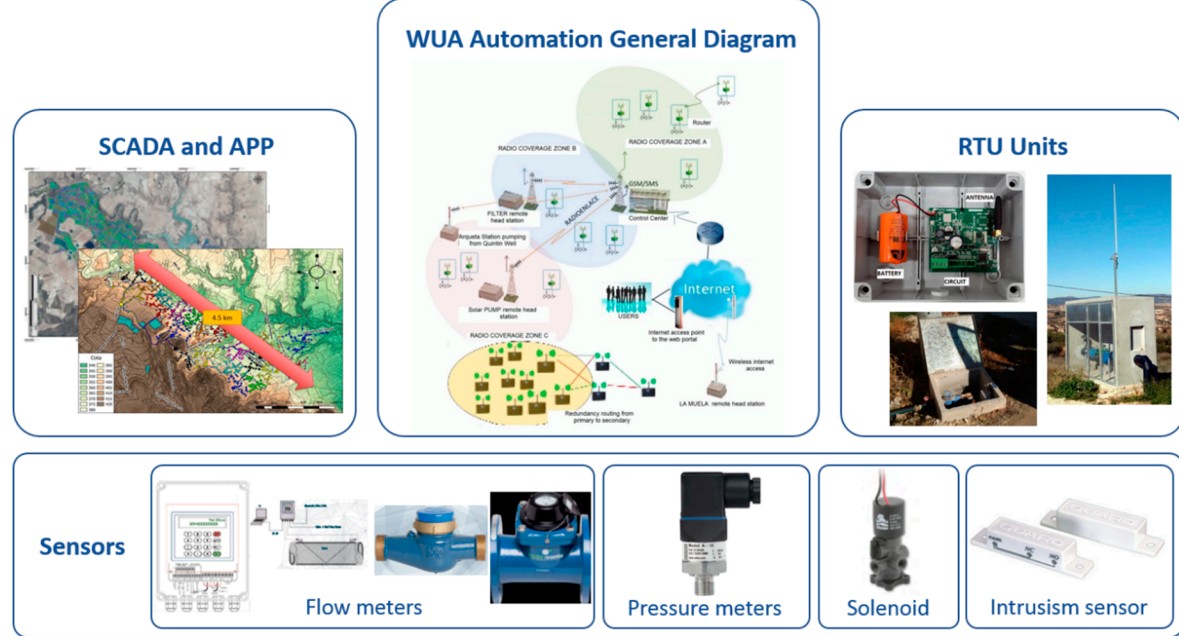

**Figure 2.** Scheme of automation system in Pliego water users association (WUA).

We focused on the development of new hardware and software expressly designed to solve communication problems and facilitate the management of irrigation schedules by users and make it easier for administrators to control the entire system with sufficient information to make decisions that are most beneficial, and for this it was necessary to satisfy three main functions:

1. System consultation: serves the managers to be able to control the operation of the system in real time (not open hydrants flow consumed, pressure at control points, etc.).
2. Acquisition of historical data: allows administrators and users to know and consult the data already recorded (consumption, pressure, valve opening/closing states, etc.).
3. Reading or on-demand action: provides the last recorded data and allows interactions with connected elements.

To do this, a common platform was designed based on a commercial application programming interface (API) that allowed a network of different systems. The irrigation system includes several levels of devices and sensors.

Taking into account these requirements, our system was designed, and here we describe the main integrated devices in this automation system, as indicated in Figure 3. There are two automation levels apart from the screen system for consulting and management (SCADA, apps, others). The first

level consists of remote transmission units (RTUs). First, in the RTUs, the concentrators can be distinguished. This device is a concentrator of communication and RTUs with inputs and outputs. Moreover, it can communicate with a central server using GPRS technology or free-band radio, and can operate continuously for 6 months in the absence of communication without a loss of information. This is a totally autonomous device. The concentrators are powered by batteries and, as additional support, can manage their charge through small solar panels. Additionally, these devices include the possibility of wireless firmware reprogramming. The most basic equipment can control up to 4 hydrants, 1 digital input and 1 output, and 2 analog inputs, although this number can be expanded by using input/output (I/O) expansions. They can work as a GPRS or radio endpoint and as a GPRS/radio mixed communications hub. These devices collect and concentrate the communications from a radio subnet and retransmit them via GPRS.

The other RTU type is the slave. These devices can communicate with a central server using GPRS technology or free-band radio, and can work continuously for 6 months in the absence of communications without a loss of information. These devices are fully autonomous. They are powered by a single lithium battery that gives them autonomy greater than 3 years in the GPRS version and 10 years in the radio version (24 daily communications). They can control hydrants and digital input, and can work as GPRS endpoints or radio. In this case, there is also the possibility of wireless firmware reprogramming. The consumption is 35 uA in the absence of communications, while consumption increases depending on the number of communications per hour and their duration (Table 1) [35,36].

**Table 1.** Annual battery consumption depending on number of communications.

| | | Number of Daily Communications | | | | | | | Always Connected (1 Hourly Communication) |
|---|---|---|---|---|---|---|---|---|---|
| | | 1 | 2 | 4 | 6 | 8 | 12 | 24 | |
| | 60 | 7.85 | 7.64 | 7.44 | 7.25 | 7.44 | 7.25 | 6.73 | 0.11 |
| | 30 | 4.37 | 4.31 | 4.24 | 4.18 | 4.24 | 4.18 | 4.00 | 0.11 |
| Analog input reading interval 4–20 mA (min) | 15 | 2.32 | 2.30 | 2.28 | 2.26 | 2.28 | 2.26 | 2.21 | 0.10 |
| | 10 | 1.58 | 1.57 | 1.56 | 1.55 | 1.56 | 1.55 | 1.53 | 0.10 |
| | 5 | 1.21 | 1.21 | 1.20 | 1.20 | 1.20 | 1.20 | 1.18 | 0.10 |

The RTUs include antennas to avoid the lack of coverage because of the topology. Hubs have antennas on 4 m masts. Concentrators simultaneously control the sector outlets.

The sensors included in this system are described as follows. Initially, 1300 units of variable caliber multiple jet counters with pulse emitters were installed. Additionally, 20 units of Woltmann meters, with a diameter of 150 mm with pulse emitters, were included. Finally, the project included 6 units of noninvasive ultrasonic flow meters. They were connected to a Siemens power line communication (PLC) in a solar pump and can also be connected to the remote units described above. The PLC is integrated in the system. The valves of the entire irrigation system are controlled by approximately 1320 units of 3-latch solenoid valves. Finally, the pressure is controlled by 28 units of 0–10 bar pressure transductors.

First, to prepare this study, data from Pliego, a village located in Murcia (Spain), and the census of farmers of Sector 1 of the WUA's Pliego network were offered in the crop inventory of the region of Murcia 2016–2017 [37]. These were analyzed in order to guide the sectors according to the type of predicted crop. The intention was to take advantage of different hydraulic needs according to the months of the year when the crop grows.

Second, it should be noted that the installation of a remote control system in the WUA of Pliego was a challenge, both technically and economically. The main characteristics that were taken into account will be described here. To prepare this study, data on water consumption of different hydrants were analyzed. This analysis was developed in order to know the current demand depending on the type of crop on the ground, based on the census of the Sector 1 network. For this purpose, invoices of each micro-plot for the last 5 years were studied. Moreover, the existing main networks that

can sectorize and telecommand from the network and the different water resources were studied (wells, reservoirs, regenerated water from wastewater treatment plants [38], pumping station systems). The data they supply to the system (flows, water quality parameters, and network pressure) every hour for mono-hydrant flow meters and in real time at critical control points of the primary network and some of the secondary network are analyzed to keep pressure levels at a value around 25 m of the water column.

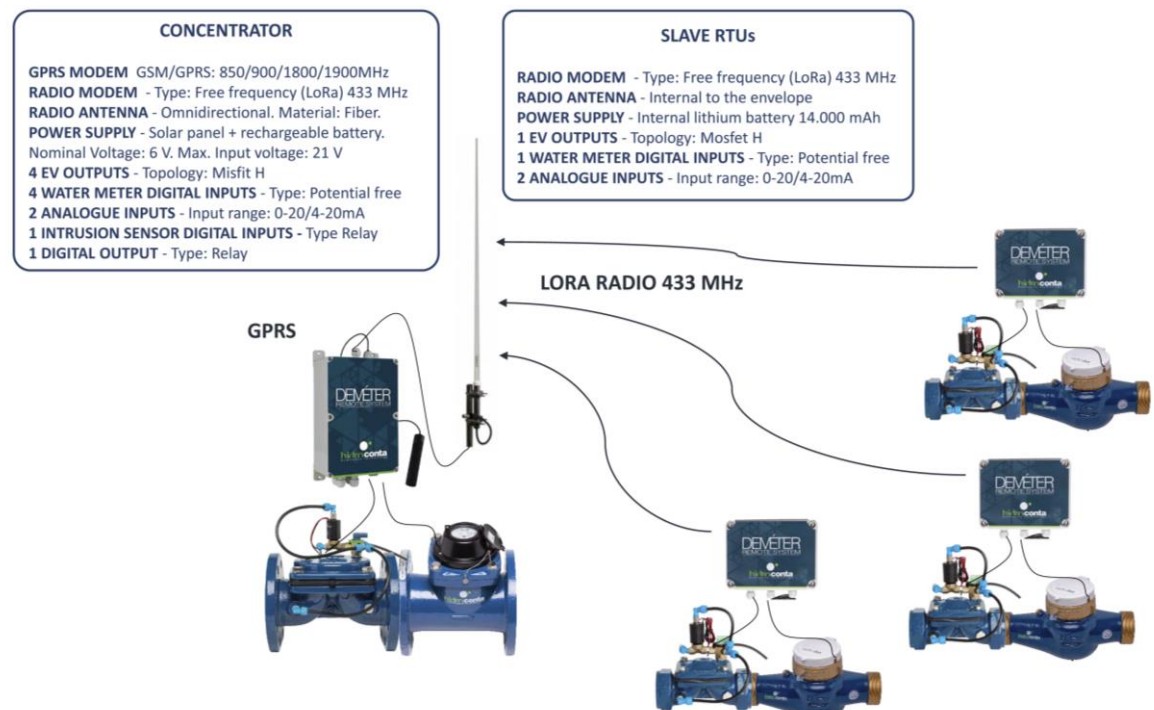

**Figure 3.** Descriptions of sensors in Pliego WUA.

Several challenges had to be solved, mainly automating an existing network and offering users the ability to select irrigation scheduling for their micro-plots. Additionally, the location in a steep terrain involves great communication difficulties. All of this has to be managed in a way that is economically and environmentally sustainable [39]. For this purpose, the different technologies available to date [40–42] and their possible uses were evaluated, analyzing the pros and cons of each.

The challenge of this research was to design a cheap, scalable, easy-to-implement and maintain remote control system with the ability to fill the gap of existing hydraulic network topologies. For this purpose, providing solutions to any WUA regardless of its extension, topography, hydrant concentration, or coverage was intended. This system would provide a solution not only for WUAs with good conditions (extension, topography, hydrant concentration, and coverage), but other WUAs that, because of their special characteristics, have had difficulty automating with existing technologies until now. In this case, the price of each device for each user socket is EUR 298 compared to EUR 1300 that the alternative of installing a remote would cost. Regarding ease of maintenance, the biggest advantage is that they use standard batteries that are easy to replace. Finally, remote control is achieved thanks to the functionality of the device to connect over the Internet.

## 2.1. Analysis of Smart Agri System

Pliego's WUA includes three characteristics that are a technical challenge for the installation of a remote control system that is economically acceptable [43]: the average size of plots, the distribution of hydrants, and the geophysical space in which it is located.

### 2.1.1. Average Parcel Size

The area of the sector under investigation, Huerta Alta (Sector I), reaches 373.59 ha. Taking into account that 1949 users are registered (Figure 4), they have an average parcel size of less than 0.2 ha per plot. This implies extreme atomization of the irrigation that makes this WUA an exceptional case, making it difficult to maintain and amortize equipment, whether hydraulic or electronic, to serve the area. The parcel used to draft the same was obtained from three fundamental sources:

- Census of the WUA of spread registers;
- Official mapping of the cadstre;
- Irrigation area recognized by the Segura River Basin.

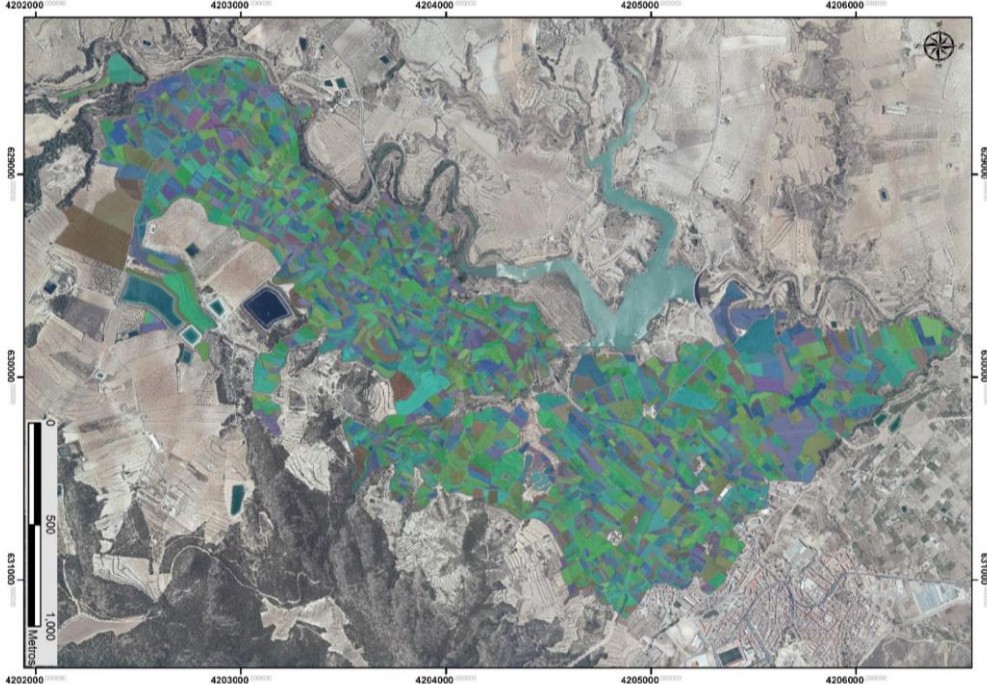

**Figure 4.** Distribution of micro-plots for remote control in Pliego WUA.

### 2.1.2. Distribution of Hydrants

The main drawback at this point is the dispersion of hydrants. The pre-existing network consists of mono-hydrant arches distributed in each plot, as opposed to the more favorable distribution of multi-hydrant arches. This feature, combined with the large number of users, made it essential to install a large number of remote terminals (Figure 5).

This means one device per 0.2 ha, which requires a large economy at various levels. For the investment, equipment with very low cost and very easy installation became indispensable. This was achieved with a device that included strictly necessary inputs and outputs, allowing the reading of a counter and the opening and closing of a valve, with limited energy consumption, avoiding increasing costs of power and energy storage.

The installation was solved by a simple protocol, but also included on-site testing of communication and operation of the hardware components for all terminals (Figure 6). In terms of operation and maintenance, the system had to be robust and durable and employ an affordable communication system. This ruled out the use of GPRS communications, leading to the use of a mixed GPRS-radio system.

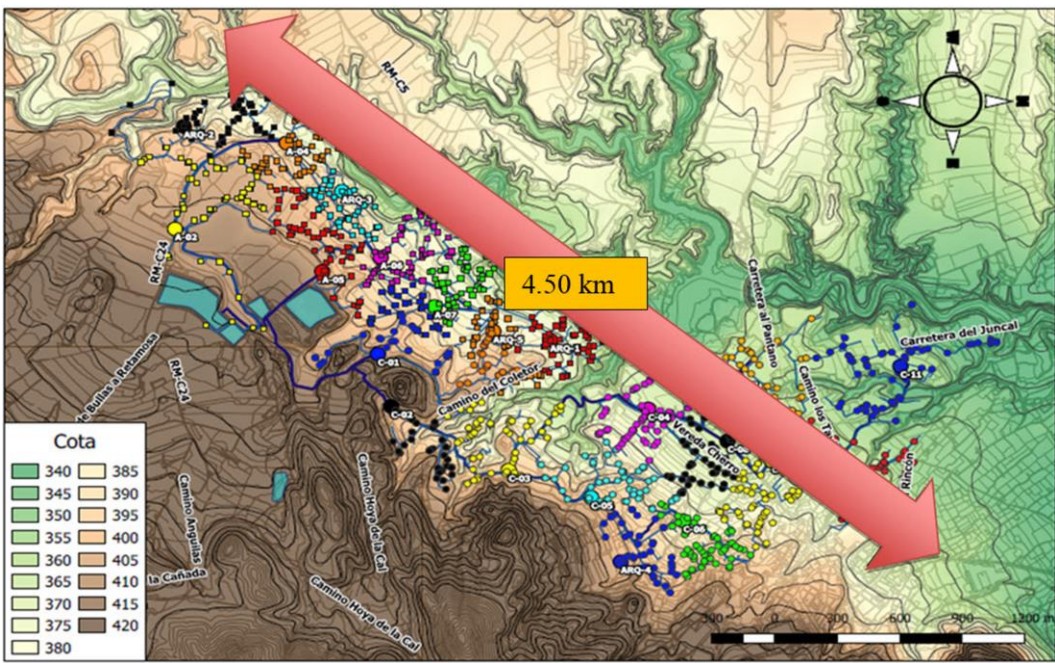

**Figure 5.** Distribution of hydrants for remote control in Pliego WUA.

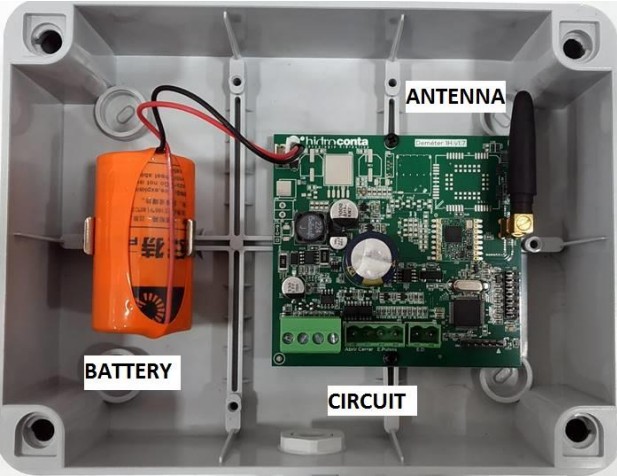

**Figure 6.** Smart Agri equipment and components (single user in a micro-plot).

For this purpose, the concentrators described above, which incorporate high-gain radio antennas that ensure the reception of communication in the ideal case of direct vision between them at distances reaching 5 km, were used (Figure 7). This distance should not be assessed in absolute terms but in relation to the size of the antennas in both the emitting and receiving equipment, in this case 270 cm in the hub and 5 cm in the slave RTU. In certain extremely difficult locations, correct communication was achieved by installing a 40 cm external antenna on the slave equipment. This was necessary in eight arches, which, considering a total of 1205 installed radio terminals (Figure 8), represent less than 0.7%. Figure 8 includes a hydrant, which consists of a flow meter with pressure regulation in order to connect the farmer's drip irrigation system.

In order to reduce equipment maintenance costs, lithium batteries with an estimated duration of 10 years were used (Figure 9). In addition, the simple installation excluded fixed anchors and unnecessary screws, allowing the possibility of moving the equipment for quick and efficient inspection and operation. The lifetime of the battery depends on different factors (accuracy of float voltage, frequency of discharges, number of discharges, maximum discharge rate, depth of discharges,

final voltage limit, operating temperature, amount of ripple current, and allowable voltage during charging and discharging). This is a strict regime that must be followed precisely to achieve the design life. Not many facilities can maintain that level of control. Temperature-wise, the standard internal temperature of the battery is 25 °C (77 °F). Battery life is cut in half for every 10 °C above 25 °C. Other factors that battery life depends on are the duty cycle (frequency with which the battery charges and discharges) and the corrosivity of the environment, among others.

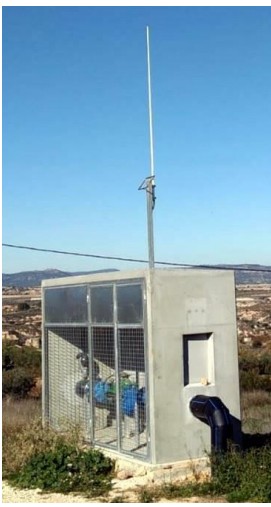

**Figure 7.** General Packet Radio Service (GPRS)-radio hub (sectorization arch).

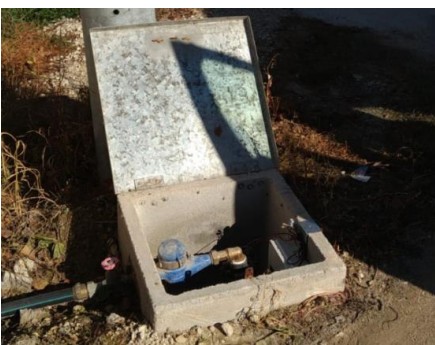

**Figure 8.** Hydrant installed in a semi-buried arch.

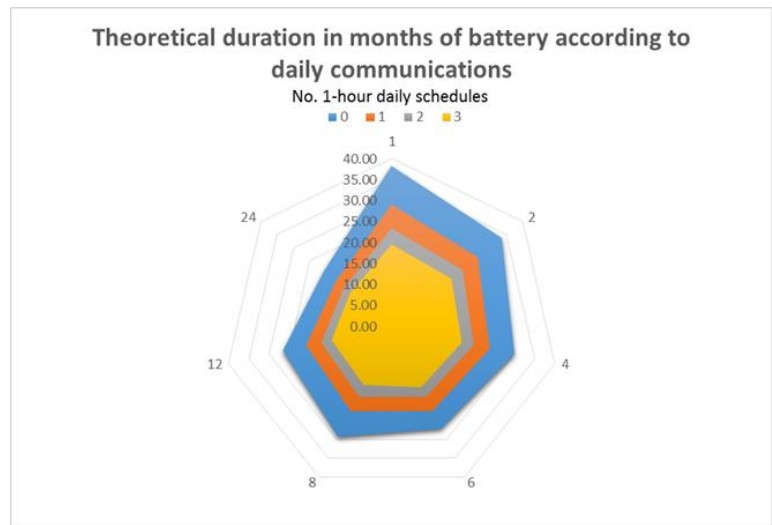

**Figure 9.** Battery duration according to number of communications.

### 2.1.3. Occupied Geophysical Space

The third feature that hinders the proper functioning of the remote control system is the natural environment in which the Huerta Alta sector is distributed. It is located on the northern slope of Sierra Espuña, in an environment with an orography in which the usual slopes are more than 1/2 and the crops reach the bed of the ravine of the Cherro (Figure 10) and others of less substance.

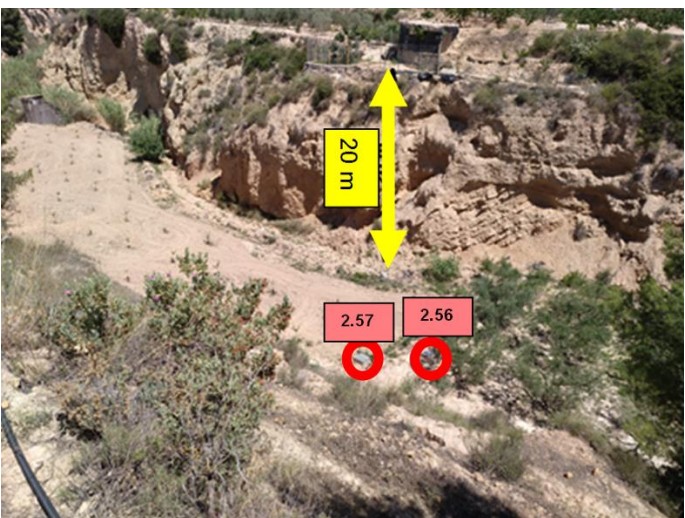

**Figure 10.** Crops in the ravine of Cherro and shots.

On the other hand, it is a highly anthropized environment, which reaches the population of Pliego and includes urbanized plots and seasonal residences. Both the orography and the existence tall buildings represent complications for the use of radio communications required by the conditions described above. To achieve the desired communication robustness, it was necessary to carry out a coverage study prior to the distribution of GPRS-radio hubs. These difficulties were enhanced by the characteristics of the arches contained in the equipment, which have very small dimensions as a result of the mono-hydrant distribution and the small caliber of the necessary hydraulic elements. This involves the emission of radio communications with the equipment installed a few centimeters from the ground, making constructions and low-rise elements possible obstacles. Another benefit of the redundancy system is the possibility to keep communications active even when there are specific obstacles such as vehicles and temporary installations.

It should be noted that given the band spectrum used and the amount of communication required per device, another aspect to consider to ensure the robustness of communication is the amount of equipment associated with each hub, which has a unique channel owner. To satisfy the amount of information to be transmitted without sacrificing the economics of operation and maintenance of the facility, a compromise solution was reached in which the number of slave radio terminals per GPRS-radio hub was limited to a maximum of 80.

## 3. Results and Discussion

As a result of the research carried out, it is worth highlighting the progress developed in the following aspects:

### 3.1. Integration of Communications in Latest Generation Free-Band

LoRa modulation is a low-power, long-range radio specification designed specifically for low-power consumption devices. Its operation, broadly speaking, is based on repeating the same message several times across its bandwidth, so that it manages, due to this sending redundancy, to overcome noise levels that would make communication with standard modulation impossible,

thus achieving sensitivity greater than −145 dBm. This characteristic translates, as mentioned above, as a significant increase in communication distance of equal power with other more traditional modulations and a high tolerance for installation in areas that by their orography do not have direct lines of antenna vision that are necessary for the proper functioning of such systems.

### 3.2. GPRS-Radio Mixed Hubs

A Smart Agri four-counter terminal can be configured to act as a hub for other terminals of one or four radio counters that route communications through it, all without losing the original ability to control the valves and counters connected to the hardware. The device, thanks to the energy support of its solar panel, can remain permanently connected to both radio and GPRS to be able to route the packets it receives from one end or the other, no matter when they are generated. This system, like other hub-based systems, has a weak point: if the hub fails, all remotes that hierarchically depend on it fail. Based on this weakness, a redundant system was developed that allows each remote terminal to identify if its primary hub is not accessible, so that the system automatically routes communications through its secondary hub. Once the main issue is fixed, the system is reset to its original configuration without user intermediation. The redundant system designed for Smart Agri provides robustness of communication similar to a solely GPRS system without the inconvenience of increasing the cost of communications, making it especially suitable for high-dispersion area checkpoints.

### 3.3. Ease of Deployment and Maintenance

For all elements of the Smart Agri system, specific firmware has been incorporated that allows the operator who is responsible for its installation to check and debug all functions of the equipment (valves, counters, analog inputs, communication, etc.). This allows people to be 100% sure that the computer is properly installed and communicating, significantly limiting installation errors. This feature is very important for deployments in hard-to-reach areas (most cases), as it avoids having to move more than once to the same location during commissioning. The hardware is designed to facilitate the maximum possible maintenance. The equipment is usually installed in places with difficult access and conditions of extreme temperature and humidity, where maintenance is not easy for long periods of time. All system elements (connectors, splice strips, etc.) have been selected so that replacement of components due to faults, firmware updates, or any other reason can be done easily and quickly and with a minimum number of tools. Batteries, for example, are standard. The installation manager can easily get them from any Internet store and they can be easily replaced with just a screwdriver.

### 3.4. Flow Detection and Alert Generation

In our specific case, the server using the Demeter system is deployed with a local server exclusive to the installation and with the location required by the client. However, it is fully compatible with a cloud server, for which it contracts commercial services from third parties, and in which different installations coexist. Privacy is secured by login and a permissions system.

The system scheme once the information reaches the server is shown in Figure 11. In this case, unlike the long-range wide-area network (LoRaWAN), there is no network server, since with GPRS hubs and terminals, the information arrives by GPRS to the application server or back end. The application server interprets the GPRS communications and payload using representational state transfer (REST) APIs. The same REST API is used by the different interfaces for access to the system or fronts (Demeter SCADA Web through any Internet browser and Hydroconta metering app through any smartphone) and by applications at the highest level of third parties with which we have integrated.

The firmware of the remote units performs an extrapolated reading of the flow passing through the counter. The equipment measures the time between detecting two pulses so that it can accurately calculate the flow where it is being watered. This information allows communities to have a deeper understanding of their networks without having to install expensive flow measurement equipment,

which also needs an external power supply in most cases. The system generates a multitude of alerts, all of them configurable by the user.

First of all, it is possible to have alerts related to the hardware, such as a communication failure, low battery, and a dirty solar panel, among others. Then, there are those related to the valves, such as closed valve flow detection and no flow detection with an open valve, and those related to the meter, such as excess flow and flow defects. Activation thresholds can be configured by the user based on their permissions. The same is true for analog-related inputs. For example, for pressure, an alarm will be generated for excess pressure or pressure defects. Thresholds are similarly configured by the user. The system also allows for configuring several phone numbers so that critical alerts can be sent via SMS and brought to the attention of responsible staff at the time they occur.

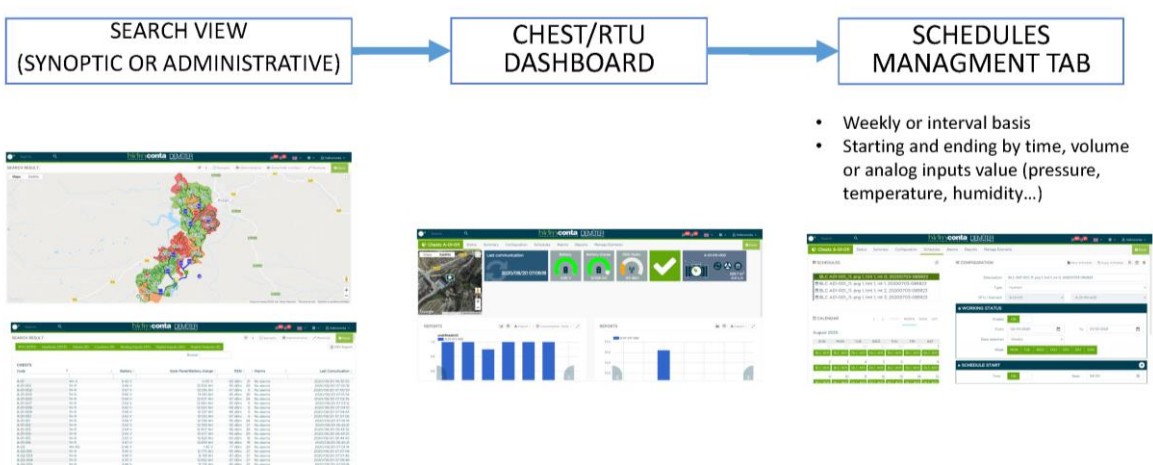

**Figure 11.** System scheme once information reaches the server.

*3.5. Elimination of Its Own Large Infrastructure*

Currently, it is very difficult to find a geographical area, especially in Spain, that lacks mobile phone coverage, which makes the situation of having to deploy a radio-only network extremely rare. In all other cases where we have equipment transmitting, either directly via GPRS or through its assigned hub to the Internet, it becomes unnecessary to install at the headquarters of the watering community any equipment related to the remote system (Figure 12).

The option is given to install the system server in the cloud, which prevents the client from maintaining the server (cooling, replicating databases, power consumption, etc.) and avoids all associated costs. Three alternatives were proposed with their specific costs, including 20 shots by sector and 1453 user shots.

The first alternative (the one presented in this paper) consists of a low-frequency communication system (LoRa). This alternative includes 21 hubs and 1453 slave RTUs. The total cost of the installation is EUR 446,644 and the annual operating cost is EUR 10,640.45. It includes replacing the autonomous battery every 10 years and replacing and maintaining equipment and GPRS communications.

The second alternative would include a traditional radio communication system with two radio links and 1473 RTUs. The installation cost in this case is EUR 981,450 and the annual operating cost is EUR 49,057.45. It includes replacing the autonomous battery every 10 years and replacing and maintaining the equipment.

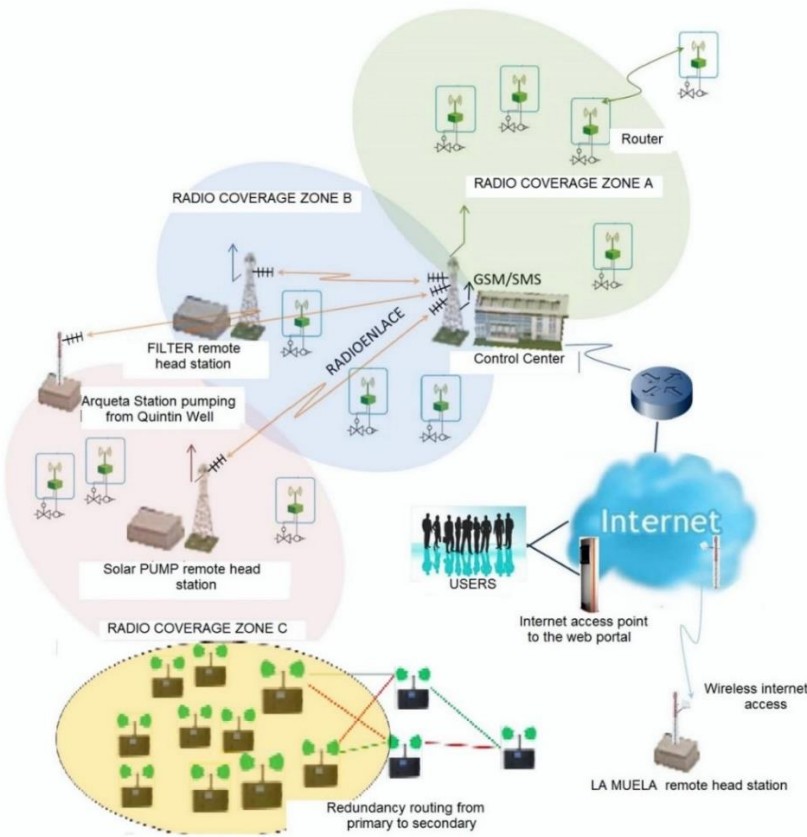

**Figure 12.** Integration scheme of different systems.

The third alternative is a complete GPRS system with 1473 RTUs, at a total installation cost of EUR 438,954 and total annual operating cost of EUR 62,925.45. It includes battery replacement every 10 years, and replacement and maintenance of GPRS equipment and communications. The details of obtaining these costs are listed in Table 2.

**Table 2.** Details of the different systems.

| Systems | Description | Amount |
|---|---|---|
| Radio long-range (LoRa) | Battery replacement (autonomous, 10 years) | 145.3 |
| | Equipment replenishment and maintenance (2%) ** | 30 |
| | GPRS communications | 21 |
| Traditional radio | Battery replacement (autonomous, 10 years) | 147.3 |
| | Equipment replenishment and maintenance (5%) ** | 74 |
| GPRS | Battery replacement (autonomous, 10 years) | 147.3 |
| | Equipment replenishment and maintenance (5%) ** | 30 |
| | GPRS communications | 1473 |

** A lower annual equipment replenishment rate (2% vs. 5%) is expected for the LoRa radio system because it is low-power radio with far fewer high-temperature conditions in field installation versus traditional high-power radio and GPRS.

To determine the viability of the system, areas planted according to crop were studied (Figure 12), and peach trees, with a higher percentage of implantation and average annual benefit estimated at EUR 3722/ha [44], were selected as a representative crop, with this value analyzed and compared. The cost–benefit graph shows the investment according to the communication system and is easily appreciated for small plots with surface values of less than 0.05 ha, as in our case (Figure 13). The other solutions require a higher rate of return and can make investment in automation unfeasible.

Despite having a similar implementation cost, the expenses and maintenance costs are higher, and this is compounded by a lack of guaranteed communication coverage, which makes them unfeasible.

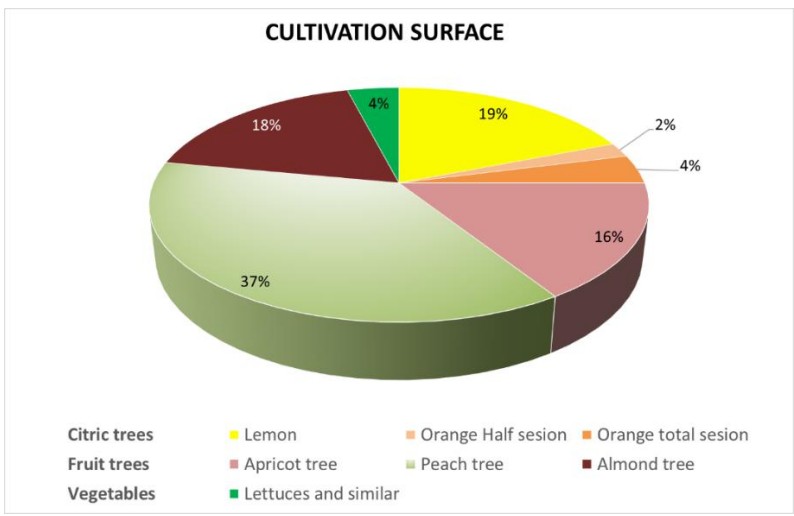

**Figure 13.** Crop type according to surface.

On the other side of the scale, the use of LoRa is evaluated, because despite being a well-known technology, it has not been satisfactory due to limitations in the communication band, which have been corrected by software that establishes communication motives every hour, with a redundancy system to guarantee the robustness of the system without depending on the weather and coverage of the area. The proposed alternative is clearly the most appropriate from the cost–benefit analysis point of view (Figure 14).

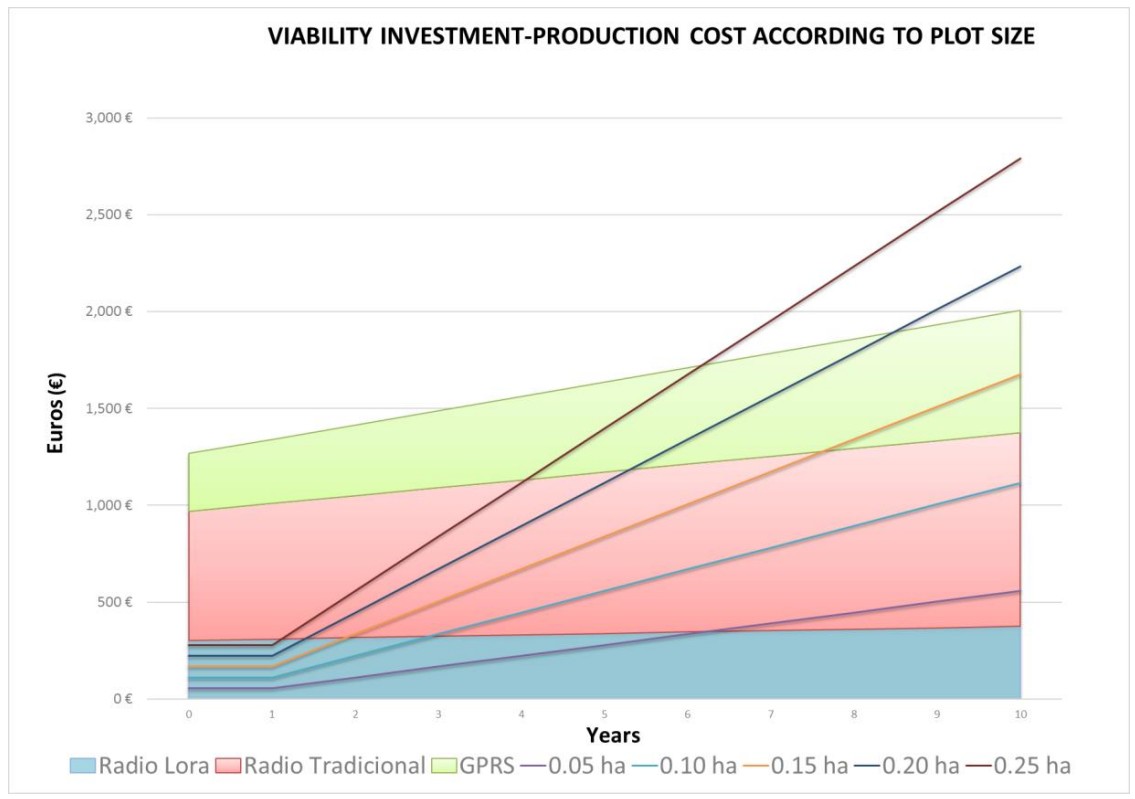

**Figure 14.** Feasibility of investment as a function of the type of architecture versus plot surface.

## 4. Conclusions

The new technology developed by Smart Agri offers the ability to control the irrigation status of crops from any point. Thanks to the modernization of irrigation in the southeast of Spain, software is being developed that improves the way farmers work. New IoT applications are preventing the abandonment of fields that have survived since time immemorial in the Segura River Basin. With these new tools, it is possible to improve the gaps; Internet agriculture, which continues to evolve, also collaborates to counter climate change, reducing the water footprint and carbon footprint. On the other hand, it is notable that situations such as the one we are currently experiencing (COVID-19 pandemic) [45] enable safer management and subsistence of vital agricultural products for the population, reducing the exposure of farmers who use these IoT technologies, even in extreme situations such as isolation.

Some of the advantages of this device are as follows:

- The telemetry solution is open and allows connection to various sensors with standardized outputs on the market.
- It is possible to monitor the state of other devices, relay outputs, and analog or digital outputs.
- The included I/O device is a data server that responds to requests from a data cloud platform or other devices such as mobile phones and tablets, among others.
- The monitoring devices are connected to the Internet. Machine-to-machine (M2M) protocols determine whether they operate independently or as part of the SCADA system. This allows communication with other devices or machines.
- Communication between devices and people is a strong point. It is possible to connect with a single click from anywhere. Additionally, it is possible to configure alerts.
- The stored data (historical and real time) are exposed on dashboards accessible by the web on any browser and are very intuitive to interpret.
- The precision agriculture telemetry platform enables graphics to be customized to the needs of the user.
- The charts are helpful to interpret what is happening and allow the user to establish better criteria for managing the farm as a whole.
- The data make agriculture smarter by improving management from the agronomic, environmental, and economic points of view.

**Author Contributions:** Conceptualization, J.C.-Z.; validation, A.R.-C. and J.M.M.-M.; supervision: A.R.-C. and J.M.M.-M.; investigation, writing—original draft preparation, writing—review and editing, conceptualization, software, formal analysis, resources, data creation, visualization, project administration: J.C.-Z., D.P.-B., and C.A. All authors read and agreed to the published version of the manuscript.

**Funding:** I appreciate the opportunity the ALICE Project "Accelerate innovation in urban wastewater management for climate change", proposal number: 734560, funded by the European Commission, Horizon 2020 Grant under the European Commission program Marie Sklodowska Curie Actions, Research and Innovation Staff Exchange (RISE) has given me to broaden my vision initially focused on the water–energy binomial and give it a global dimension. It has allowed me to seek integration through IoT technologies that bring technological knowledge and distance to people with software that is currently being developed and allows remote control of crops to reduce the consumption of water and energy, reduce pollution, mitigate climate change, and improve the use of recovered water applied to agriculture.

**Acknowledgments:** Our gratitude goes to the Hidroconta Company, Miguel Hernández University, the Region of Murcia, the participating water user's association of Pliego, and especially Miguel Angel del Amor Saavedra, since without his contribution it would have been impossible to carry out this work.

**Conflicts of Interest:** The authors declare no conflict of interest.

## Nomenclature

| | |
|---|---|
| API | Application programming interface |
| BBDD | Databases |
| FSK | Frequency shift keying |
| GPRS | General Packet Radio Service |
| GSM | Global System for Mobile Communications |
| ICT | Information and communication technology |
| LoRad | Low radiation of electric and magnetic alternating fields |
| NBIoT | Narrowband IoT |
| RTU | Remote terminal unit |
| SASW | Smart Agri SCADA web |
| SCADA | Supervisory Control and Data Acquisition |
| SDGs | Sustainable Development Goals |
| SMS | Short message service |
| TST | Tajo-Segura transfer |
| WUA | Water user's association |
| WWTP | Wastewater treatment plant |

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
