# Peer review of "Adaptation of a Traditional Irrigation System of Micro-Plots to Smart Agri Development: A Case Study in Murcia (Spain)"

_agronomy, doi:10.3390/agronomy10091365_

Round 1

Reviewer 1 Report

This is a very good and interesting paper.

I have only one comment:

This research was based on a 'very small' plot: 0.2 ha.

May be the authors can say something related to the size of the plot. Especially if they are significant larger than 0.2 ha. Or can you simply expect the similar results?

In my opinion this is a very useful and applicable research!

Author Response

Dear reviewer,

All the comments that you asked us, has been included in the next file.

Thanks for your collaboration.

A general revision of the English language in the article has been saked to the editorial team.

Best regards. 

Reviewer 2 Report

See attached file

Author Response

Dear reviewer,

We are included all of your observations.

A general review of the English version has been in charge to the editorial.

Best regards.

Round 2

Reviewer 2 Report

Dear authors,

I appreciate the revision you have implemented.

I have a couple of comments I am sure you can do without any important effort.

  • References 11 and 14 are still written in Spanish.

11. Naciones unidas. Objetivos de desarrollo sostenible: Objetivo 15.3

14. Naciones unidas. Objetivos de desarrollo sostenible: Objetivo 12.4

  • In line 65, you indicate "families only have weekends to pay attention". For me these details are significant. Some users will pay attention on weekends or Mondays. The important is this technology can allow them to devote not too much time.

Regards

Author Response

Dear reviewer,

The last changes has been included.

Best regards.
